# CVPT: Cross-Attention help Visual Prompt Tuning adapt visual task

## Abstract

In recent years, the rapid expansion of model sizes has led to large-scale pre-trained models demonstrating remarkable capabilities. Consequently, there has been a trend towards increasing the scale of models. However, this trend introduces significant challenges, including substantial computational costs of training and transfer to downstream tasks. To address these issues, Parameter-Efficient Fine-Tuning (PEFT) methods have been introduced. These methods optimize large-scale pre-trained models for specific tasks by fine-tuning a select group of parameters. Among these PEFT methods, adapter-based and prompt-based methods are the primary techniques. Specifically, in the field of visual fine-tuning, adapters gain prominence over prompts because of the latter's relatively weaker performance and efficiency. Under the circumstances, we refine the widely-used Visual Prompt Tuning (VPT) method, proposing Cross Visual Prompt Tuning (CVPT). CVPT calculates cross-attention between the prompt tokens and the embedded tokens, which allows us to compute the semantic relationship between them and conduct the fine-tuning of models exactly to adapt visual tasks better. Furthermore, we introduce the weight-sharing mechanism to initialize the parameters of cross-attention, which avoids massive learnable parameters from cross-attention and enhances the representative capability of cross-attention. We conduct comprehensive testing across 25 datasets and the result indicates that CVPT significantly improves VPT's performance and efficiency in visual tasks. For example, on the VTAB-1K benchmark, CVPT outperforms VPT over 4% in average accuracy, rivaling the advanced adapter-based methods in performance and efficiency. Our experiments confirm that prompt-based methods can achieve exceptional results in visual fine-tuning.

## 1 Introduction

Increasing the scale of the models is a common method to enhance the model's performance (35)(9)(28)(29). In recent years, with the rapid development of computing devices, model sizes have significantly increased (45)(6)(16)(47). For instance, the number of parameters in the GPT series developed by OpenAI has surged from 117 million to 1.8 trillion in just five years (36)(37)(2). The rapidly increasing number of parameters will lead to the problem of immense computational overhead. Therefore, adapting those models to downstream tasks with the full-tuning method will incur enormous costs. To resolve this issue, the PEFT approach has been proposed (19)(27)(1)(38)(5). PEFT adapts those large-scale pre-trained models to downstream tasks in a more efficient way by fine-tuning a subset of the models that contains much fewer parameters. Two mainstream methods within PEFT are Adapter (18) and Prompt (27). During the training process, the Adapter inserts adapters into each transformer block and tunes those adapters, while the Prompt inserts prompt tokens into the embedded tokens to update the prompt tokens.

VPT, a prompt-based method is first introduced by Jia *et al.* (21) for visual fine-tuning tasks. Nevertheless, research on the adapter-based method is prominent due to its superior performance. Although some works have improved the performance of VPT (20)(12)(7), it is still challenging to match the effectiveness to that of adapter-based methods. There appears to be a consensus that prompt-based methods underperform adapter-based methods in the visual domain. But is this the case?

We conduct extensive experiments and analyses on VPT to uncover the reasons for its weaker performance compared to the Adapter. According to our experiments, we consider that the primary reason for the performance difference between VPT and adapters is that VPT's deployment directly applies that used in NLP tasks (27), without any adaptation to visual tasks. In NLP tasks, prompts usually contain rich semantic information that guides the fine-tuning process of the model. However, in visual tasks, prompts lack representation information. Therefore, it is necessary for VPT to use an abundant amount of prompts to fine-tune models. However, the design of VPT leads to computational inefficiency and redundancy, as well as the disruption of the self-attention between embedded tokens 3.1. As the graph follows 1, VPT shows a significant decrease in performance and an increase in costs when given a large number of prompts. Considering that, we think that **VPT is unusable when given a large number of prompts.**

To handle the problem, we redesign VPT and introduced Cross Visual Prompt Tuning (CVPT). For the prompt tokens in CVPT, we calculate the cross-attention with the embedded tokens and add the result as residuals to the embedded tokens. This approach avoids the computational complexity of self-attention that is quadratically related to the number of prompts and allows prompts to focus on the embedded token to adapt to downstream tasks more efficiently. Additionally, by maintaining consistency in token dimensions throughout the computation process, the results of cross-attention can be directly summed with embedded tokens as residuals and do not introduce additional computational overhead for subsequent MLP. Furthermore, we share the weights of the self-attention layer with the cross-attention layer during loading checkpoints, keeping the cross-attention layer frozen alongside the self-attention layer, which eliminates the requirement for additional learned parameters for the cross-attention, and utilizes the encoded information in self-attention to help the fine-tuning of the model.

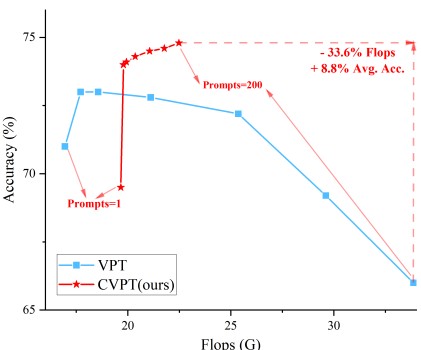

Figure 1: **Comparisons of performance and Flops between VPT and our CVPT** with a pre-trained ViT-B/16 model on the VTAB-1k benchmark. We set the number of prompts to 1,10,20,50,100,150,200 respectively.

We validate the effectiveness of our method on 25 datasets, the results show that the CVPT achieves a significant improvement in performance and efficiency compared to the VPT. CVPT shows an average **4%** improvement in accuracy on the 19 VTAB-1K datasets, **1%** on the 5 FGVC datasets, and **3%** on the ADE20K dataset. Additionally, if given fewer prompt tokens, CVPT achieves a comparable performance with other advanced PEFT methods which significantly outperforms the other prompt-based methods and needs fewer learnable parameters. If a large number of prompts is allowed, our CVPT outperforms the SOTA methods on FGVC and ADE20K datasets. Besides, although a large number of prompts are inserted, it does not introduce too much extra computational overhead compared to VPT.

Finally, we explore the impact of the deployment's position and the effectiveness of the weight-sharing mechanism. The improvement on the model can be fully illustrated by the experimental results above, indicating that prompt-based methods can also rival SOTA adapter-based methods.

Overall, our contributions are as follows:

- We provide a detailed analysis of the application of VPT to visual tasks, and propose that its drawback can be summarised in three points which are **lack of adaptation**, **computational inefficiency and redundancy**, **destruction of self-attention**.
- We propose CVPT, which introduces cross-attention and weight-sharing mechanisms, to avoid the efficiency and performance problems caused by VPT, which allows us to use more prompts to improve performance efficiently.

- We conducted experiments on 25 datasets with different downstream tasks. The results show that our approach significantly outperforms the original VPT and other prompt-based works in terms of performance and efficiency. It is also comparable to SOTA adapter-based methods, demonstrating the usability of the prompt-based approach for visual fine-tuning.

## 2 Related Work

**PEFT.** In the era of CNN, making bigger and deeper models was an effective way to improve performance (26)(15)(43). With the rise of transformers, this trend became even more popular. The introduction of ChatGPT further cemented the goal of the community to develop larger and more powerful models. However, limited by their scale, despite their powerful performance and generality, these large models are difficult to adapt downstream tasks by using traditional paradigms (full-tuning). Consequently, NLP researchers first proposed PEFT methods. Their works demonstrate that fine-tuning just a small number of parameters in a large-scale pre-trained model can achieve nearly the same performance as full-tuning. Encouraged by the success in NLP, researchers began to apply PEFT to large-scale vision models on different visual tasks (8)(44). After development in the past several years, the mainstream PEFT methods can be broadly categorized into adapter-based methods and Prompt-based methods.

**Adapter.** Jie *et al.* (18) proposed inserting adapters into the network to efficiently fine-tune the model. These adapters are commonly a small network that usually contains an upsampling layer and a downsampling layer. The input is multiplied with a scaling factor after passing through the upsampling and downsampling layers and then the result is added as a residual to the input. The general form of adapter can be expressed as:

$$X_{out} = X_{in} + \gamma(W_{up}(W_{down}(X_{in}))),$$ (1)

where $X_{in}$ denotes the input of Adapter, $\gamma$ represents the scaling factor of Adapter, and $W_{up}$ and $W_{down}$ correspond to the upsampling layer and downsampling layer, respectively. Some works did some adaption to visual tasks based on Adapter, developing several variants such as AdaptFormer (4), LoRA (19) and RepAdapter (30), *etc*. These adapter-based methods dominate the field of visual fine-tuning.

**Prompt.** Prompt was originally used in the field of NLP which is added to the input text for comprehension tasks. Lester *et al.* (27) proposed treating the prompt as a continuous vector and fine-tuning the model by updating its gradients. Jia *et al.* (21) introduced this concept to visual fine-tuning for the first time, naming it VPT. As shown in Fig.3, the embedded tokens are spliced with the prompt tokens before entering each transformer block, allowing it to participate in every layer of the network within the transformer block. Before entering the next transformer block, the prompt tokens of the previous layer are discarded, and new prompt tokens are spliced with the embedded token again (VPT-Deep). This can be formulated as shown below:

$$[\vec{x}_i, \_\_, \vec{E}_i] = L_i([\vec{x}_{i-1}, \vec{P}_{i-1}, \vec{E}_{i-1}]),$$ (2)

where the red and blue indicate learnable and frozen parameters, respectively. $P$ denotes a learnable d-dimensional vector, X is the CLS token, and E is the patched image. Although there are improved variants based on VPT, such as E2VPT (12), EXPRESS (7) and DAM-VP (20), a performance gap remains between prompt-based and adapter-based approaches.

## 3 Method

### 3.1 Analysis of previous VPT

Firstly, we analyze VPT deeply to explore why it is not better than adapter in terms of performance and efficiency, our analysis follows three points:

**Lack of adaptation to visual tasks.** In NLP, each token represents an actual word with rich semantic information. Therefore, the processing of concatenating prompt tokens and embedded tokens is natural and suitable for NLP tasks. However, in visual tasks, tokens represent image patches and contain sparse semantic information compared to those in NLP. Therefore, simply splicing the prompt tokens with the embedded tokens may not provide sufficient guidance information. Additionally,

visual tasks often require a deeper understanding of spatial relationships and structural features of an image, which are difficult to achieve with prompt tokens.

**Computational inefficiency and redundancy.** When computing self-attention, the attention between each token and all other tokens needs to be calculated. Its computational complexity is $n^2$, where $n$ is the number of embedded tokens. If $m$ represents the number of inserted prompt tokens, the computational complexity of self-attention in VPT can be expressed as $(n + m)^2$. This increases the computational overhead significantly, especially when using a larger number of prompt tokens. Additionally, we found that prompt tokens are involved in the MLP computation process, which not only adds computational overhead but also does not impact the results. Our experiments show that removing the prompt token after self-attention does not affect the results.

**Destruction of self-attention between embedded tokens.** After softmax, the sum of the weights of all tokens is normalized to 1. Whereas, due to the addition of the prompt tokens, the sum of the weights of the embedded tokens is reduced by the prompt tokens, which corresponds to the weakening of the representation ability of the self-attention between embedded tokens. Since the prompt token is eventually removed, this is equivalent to multiplying the self-attention result between the embedded tokens by a factor which less than one. To explore how large this effect is, we set the number of prompts to 1,5,20,50,100,150,196 respectively, and visualize the tensor after the softmax function, the results are shown in Fig.2 below.

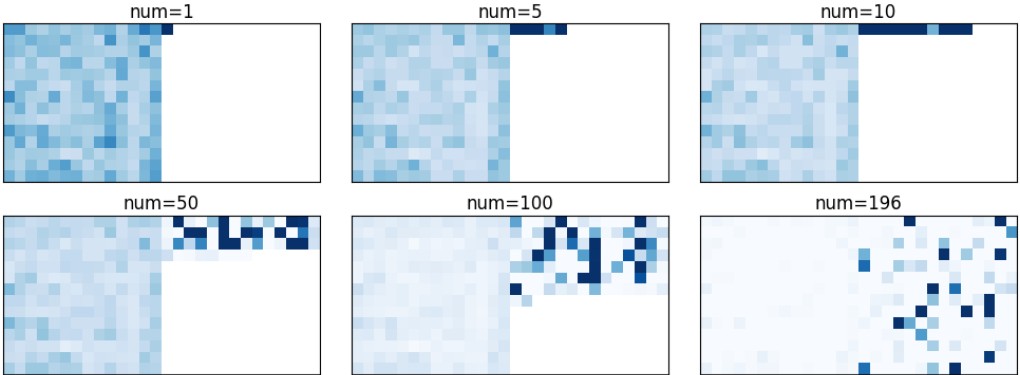

Figure 2: **Self-attention weight obtained by prompt tokens and embedded tokens.** We visualize the self-attention of $cls_{token}$ and exclude itself to observe the attention of $cls_{token}$ to other tokens. And the darker the color, the larger the weight. When giving 196 prompts, the attention weight obtained by prompts is over 80%, which greatly influences the self-attention received by embedded tokens.

As the number of prompts increases, the sum of the prompt's weight values exceeds 0.8, which is over 4 times that of embedded tokens, significantly disrupting the self-attention between the embedded tokens. This explains why VPT performance decreases substantially with a larger number of prompts.

## 3.2   Cross Visual Prompt Tuning

**Cross-Attention.** Unlike self-attention (40), which computes the relationship between each element in the input sequence, cross-attention computes attention on two different sequences to process the semantic relationship between them (3). For example, in translation tasks, cross-attention is used to compute the attention weights between the source language sentence and the target language sentence. In our method, we introduce cross-attention to handle the semantic relationship between embedded tokens and prompt tokens, guiding the fine-tuning of the model. Specifically, the input of cross-attention consists of two parts: $X_1$ and $X_2$, in which $X_1 \in \mathbb{R}^{n \times d_1}$ and $X_2 \in \mathbb{R}^{m \times d_2}$. And $X_1$ serves as the query set and $X_2$ serves as the key-value set. We set $Q = X_1 W^Q$ and $K = V = X_2 W^K$, and then the cross-attention can be expressed as follows:

$$CrossAttention(X_1, X_2) = Softmax\left(\frac{Q \cdot K}{\sqrt{d_k}}\right) V. \tag{3}$$

In which $W^Q \in \mathbb{R}^{d_1 \times d_k}$ and $W^K \in \mathbb{R}^{d_2 \times d_k}$ are learned projection matrix, $d_k$ is the dimension of value-key set. In our methods, $d_1 = d_2 = d_k$. And the shape of output is $n \times d_k$, which is consistent with $X_1$.

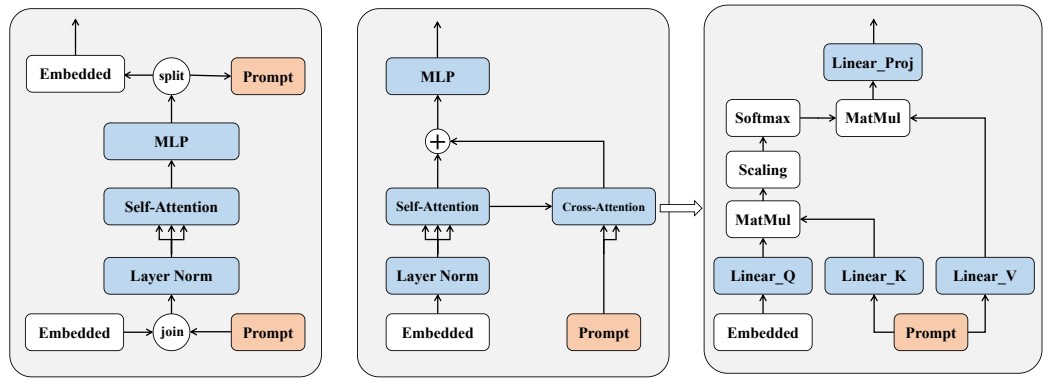

**Visual Prompt Tuning (VPT)**  **Cross Visual Prompt Tuning (CVPT)**

Figure 3: **Structure comparison of VPT and CVPT.** In which blue represents frozen parameters and orange represents learnable parameters.

**Cross Visual Prompt Tuning.** We redesign the prompt to better adapt visual tasks and proposed CVPT. Our approach, as illustrated in Fig.3, follows the VPT, the main parameters of the network remain frozen, and only the final classification layer and the prompt are trainable. The key difference is that we allow the prompt token to perform cross-attention with the embedded tokens and the result of cross-attention is added with the embedded tokens as residuals. This operation helps prompts adapt visual tasks a lot, and we demonstrate how significant this improvement is in Sec.4.2. Specifically, for any input $x_i$ of a transformer block, the forward flow can be represented as follows:

$$X_1 = X_i + SA(LN_1(X_i)), \tag{4}$$
$$X_2 = X_1 + CA(X_1, Prompt), \tag{5}$$
$$X_{out} = X_2 + MLP(LN_2(X_2)), \tag{6}$$

where blue denotes frozen parameters and red denotes trainable parameters, SA denotes self-attention, CA denotes cross-attention, and LN denotes layer normalization.

In CVPT, we only introduce linear computational overhead associated with the number of prompt tokens. It allows CVPT to use a large number of prompt tokens to improve its performance by introducing an acceptable overhead. Furthermore, CVPT preserves the original procedure of self-attention, keeping the complete representation ability of embedded tokens. We demonstrate the improvement over VPT in terms of performance and efficiency in Sec.3.3. Finally, we set embedded tokens as query set and prompt tokens as key-value set, so that we can maintain the unity of the number of channels, allowing the result of cross-attention to be directly summed with the input as a residual term.

**Weight-sharing mechanism.** The utilization of cross-attention, which requires a large number of learnable parameters (usually $\geq 30\%$ model's parameter number), leads to a major challenge in computational overhead. Therefore, if the parameters of them are tunable, the computational overhead of CVPT will even rival those using full-tuning. Therefore, we introduce the weight-sharing mechanism. Due to the structure of cross-attention equals to that of self-attention, we consider that the weight of self-attention is also instructive for the fine-tuning of cross-attention. Thus, we initialize the weight of cross-attention with the parameters of self-attention when loading checkpoints. It avoids the introduction of a huge number of learnable parameters in cross-attention and keeps the efficiency of our CVPT. We explore the impact of weight-sharing in 4.3 and demonstrate that frozen cross-attention is even more effective than learnable cross-attention.

### 3.3 Comparison with VPT

**Performance improvement.** To investigate how much improvement CVPT makes and the effect of the number of prompts on performance, we use different numbers of prompt tokens and conduct

experiments on VTAB-1K using VPT and CVPT, respectively. The results are shown in the following Table.1:

Table 1: **Performance comparisons With VPT and CVPT on VTAB-1K benchmark of different number of prompt tokens.**

| Number
Method | 1 | 5 | 10 | 20 | 50 | 100 | 150 | 200 |
|---|---|---|---|---|---|---|---|---|
| VPT | **71.0** | 73.0 | 73.0 | 72.8 | 72.2 | 69.2 | 66.0 | 64.0 |
| CVPT | 69.5 | **73.5** | **74.0** | **74.1** | **74.3** | **74.5** | **74.6** | **74.8** |

These results show that our CVPT achieves better performance in almost every case except the number of prompts equals 1. As we analyzed in Section 3.1, VPT represents a pool absolute performance on account of the lack of adaptation to visual tasks. Besides, due to the corruption of self-attention between embedded tokens, when given a larger number of prompt tokens, VPT shows significant performance degradation or even crashes. In contrast, our CVPT avoids suffering from these problems. Additionally, its performance improves as the number of prompt tokens increases. All these results above indicate that cross-attention between prompt tokens and embedded tokens helps prompts adapting the visual tasks and instruct the model's fine-tuning more exactly.

**Efficiency improvement.** To explore the improvement in efficiency of CVPT, we also recorded the amount of GPU memory occupied by VPT and CVPT during training and testing as well as the total computation of the two when conducting the above experiments, and the results are shown in Fig.4 follows:

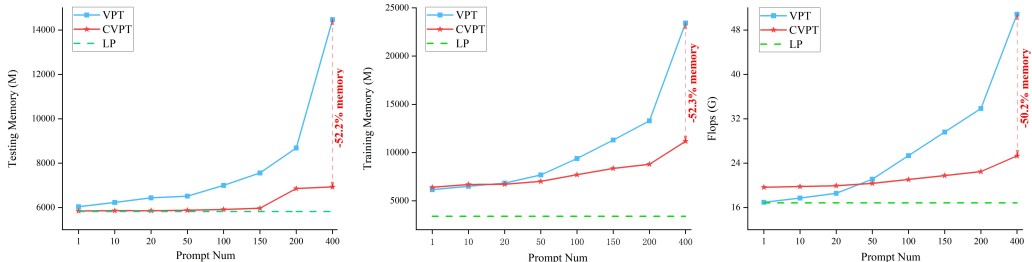

Figure 4: **The trends of training memory, testing memory, and Flops with the variation in the number of prompt tokens.** Where LP represents Linear Probing which only tunes the final classifier linear. We record those data on cifar100 in VTAB-1K, the batch_size is set to 32. Pre-trained model is ViT-B/16.

It can be seen that our CVPT has made significant improvements in efficiency compared to VPT especially given a large amount of prompt tokens. Although it requires slightly more GPU memory during testing compared to full-tuning which is marginal compared to VPT. Additionally, the weight-sharing mechanism allows for targeted optimization in engineering applications, letting cross-attention and self-attention share memory, further widening the efficiency gap with VPT. Moreover, the careful design of CVPT prevents explosive growth in memory and computation as the number of prompts increases. This means we can improve the performance of CVPT by increasing the number of prompts, which is more computationally efficient than other methods.

In summary, **our CVPT significantly improves the performance and efficiency of VPT by introducing cross-attention and the weight-sharing mechanism, especially given a larger number of prompts.** Therefore, it allows us to introduce more prompts to the prompt-based method in an efficient manner, thus improving its performance. We will demonstrate how much this improvement is and compare it with the SOTA methods in the next section.

## 4 Experiment

### 4.1 Experimental settings

**Datasets.** We evaluate our CVPT on both image classification and semantic segmentation tasks to verify its effectiveness. The specific datasets involved in our work are presented in the following.

- **VTAB-1K.** VTAB-1K comprises 19 datasets from different domains, classified into three main categories: the Natural group (natural images captured by standard cameras) (25)(32)(10)(34), the Specialized group (professional images captured by specialized equipment, such as medical and remote sensing images) (41)(17), and the Structured group (synthetic images from artificial environments). Each task contains only 1,000 training samples (22)(11)(31). This is a primary metric for evaluating PEFT's performance.

- **FGVC.** FGVC consists of five fine-grained visual classification benchmarks, including CUB-200-2011 (42), NABirds (39), Oxford Flowers (33), Stanford-Dogs (23) and Stanford-Cars (24). Unlike VTAB-1K, the datasets in FGVC benchmarks are complete.

- **ADE20K.** ADE20K (50) contains more than 25,000 images and is primarily used for scene perception, parsing, segmentation, multi-object recognition, and semantic understanding. This adaptation is challenging due to the huge gap between the objectives of pretraining and downstream tasks.

**Baseline.** We primarily use CVPT to compare with the following methods: (1) Full-tuning, (2) Adapter and its improved variants such as LoRA, Adaptformer, RepAdapter, and SPT, and (3) VPT and its variants, including E2VPT, EXPRESS and so on.

**Training.** We use the ViT-Base-16 model as our main model and AdamW as our optimizer. The other settings and training strategies follow those used in VPT. To avoid extensive hyperparameter search, we only select the number of prompts from [1, 5, 10, 20] for VTAB-1K. Besides, we use single NVIDIA 3090 on VTAB-1K and FGVC benchmark, and use NVIDIA 3090 × 8 on ADE20k.

## 4.2 Comparison with the SOTA

**VTAB-1K.** We compared our method with other baseline methods on the VTAB-1K benchmark. The experimental results are shown in Table.2, where we report the top-1 accuracy of these methods. In the table, we divide the prompt-based methods into one group and the other methods into another group. The bold values in each group represent the best accuracy.

Table 2: **Performance comparisons on the VTAB-1k benchmark with ViT-B/16 models pretrained on ImageNet-21K.**

| Method | Params. (M) | Avg. Acc. | | Natural | | | | | | | Specialized | | | | Structured | | | | | | | |
|---|---|---|---|---|---|---|---|---|---|---|---|---|---|---|---|---|---|---|---|---|---|---|
| | | | CIFAR-100 | Caltech101 | DTD | Flowers102 | Pets | SVHN | Sun397 | Patch Camelyon | EuroSAT | Resisc45 | Retinopathy | Clevr/count | Clevr/distance | DMLab | KITTI/distance | dSprites/loc | dSprites/ori | SmallNORB/azi | SmallNORB/ele |
| Full-tuning | 85.8 | 68.9 | 68.9 | 87.7 | 64.3 | 97.2 | 86.9 | 87.4 | 38.8 | 79.7 | 95.7 | 84.2 | 73.9 | 56.3 | 58.6 | 41.7 | 65.5 | 57.5 | 46.7 | 25.7 | 29.1 |
| Linear-probing (14) | 0 | 57.6 | 63.4 | 85.0 | 63.2 | 97.0 | 86.3 | 36.6 | 51.0 | 78.5 | 87.5 | 68.6 | 74.0 | 34.3 | 30.6 | 33.2 | 55.4 | 12.5 | 20.0 | 9.6 | 19.2 |
| Bias (46) | 0.10 | 65.2 | 72.8 | 87.0 | 59.2 | 97.5 | 85.3 | 59.9 | 51.4 | 78.7 | 91.6 | 72.9 | 69.8 | 61.5 | 55.6 | 32.4 | 55.9 | 66.6 | 40.0 | 15.7 | 25.1 |
| Adapter (18) | 0.15 | 73.9 | 69.2 | 90.1 | 68.0 | 98.8 | 89.9 | 82.8 | 54.3 | 84.0 | 94.9 | 81.9 | 75.5 | 80.9 | 65.3 | 48.6 | 78.3 | 74.8 | 48.5 | 29.9 | 41.6 |
| NOAH (48) | 0.36 | 75.5 | 69.6 | 92.7 | 70.2 | 99.1 | 90.4 | 86.1 | 53.7 | 84.4 | 95.4 | 83.9 | 75.8 | 82.8 | 68.9 | 49.9 | 81.7 | 81.8 | 48.3 | 32.8 | 44.2 |
| AdaptFormer (4) | 0.15 | 74.7 | 70.8 | 91.2 | 70.5 | 99.1 | 90.9 | 86.6 | 54.8 | 83.0 | 95.8 | 84.4 | 76.3 | 81.9 | 64.3 | 49.3 | 80.3 | 76.3 | 45.7 | 31.7 | 41.1 |
| LoRA (19) | 0.29 | 74.5 | 67.1 | 91.4 | 69.4 | 98.8 | 90.4 | 85.3 | 54.0 | 84.9 | 95.3 | 84.4 | 73.6 | 82.9 | 69.2 | 49.8 | 78.5 | 75.7 | 47.1 | 31.0 | 44.4 |
| RepAdapter (30) | 0.23 | 76.1 | 72.4 | 91.6 | 71.0 | 99.2 | 91.4 | 90.7 | 55.1 | 85.3 | 95.9 | 84.6 | 75.9 | 82.3 | 68.0 | 50.4 | 79.9 | 80.4 | 49.2 | 38.6 | 41.0 |
| SPT-Adapter (13) | 0.34 | 76.2 | 72.9 | 93.2 | 72.5 | 99.3 | 91.4 | 88.8 | 55.8 | 86.2 | 96.1 | 85.5 | 75.5 | 83.0 | 68.0 | 51.9 | 81.2 | 82.4 | 51.9 | 31.7 | 41.2 |
| SPT-LoRA (13) | 0.48 | 76.4 | 73.5 | 93.3 | 72.5 | 99.3 | 91.5 | 87.9 | 55.5 | 85.7 | 96.2 | 85.9 | 75.9 | 84.4 | 67.6 | 52.5 | 82.0 | 81.0 | 51.1 | 30.2 | 41.3 |
| VPT-shallow | 0.06 | 67.8 | 77.7 | 86.9 | 62.6 | 97.5 | 87.3 | 74.5 | 51.2 | 78.2 | 92.0 | 75.6 | 72.9 | 50.5 | 58.6 | 40.5 | 67.1 | 68.7 | 36.1 | 20.2 | 34.1 |
| VPT-Deep (21) | 0.53 | 72.0 | 78.8 | 90.8 | 65.8 | 98.0 | 88.3 | 78.1 | 49.6 | 81.8 | 96.1 | 83.4 | 68.4 | 68.5 | 60.0 | 46.5 | 72.8 | 73.6 | 47.9 | 32.9 | 37.8 |
| EXPRESS (7) | 0.98 | 72.9 | 78.0 | 89.6 | 68.8 | 98.7 | 88.9 | 89.1 | 51.9 | 84.8 | 96.2 | 80.9 | 74.2 | 66.5 | 60.4 | 46.5 | 7.6 | 78.0 | 49.5 | 26.1 | 35.3 |
| DAM-VP (20) | 2.52 | 73.1 | - | - | - | - | - | - | - | - | - | - | - | - | - | - | - | - | - | - | - |
| $E^2$VPT (12) | 0.27 | 73.9 | 78.6 | 89.4 | 67.8 | 98.2 | 88.5 | 85.3 | 52.3 | 87.8 | 96.1 | 84.8 | 73.6 | 71.7 | 61.2 | 47.9 | 75.8 | 80.8 | 48.1 | 31.7 | 41.9 |
| CVPT | 0.10 | 76.2 | 73.0 | 90.0 | 73.8 | 99.2 | 91.2 | 90.0 | 54.4 | 84.0 | 96.5 | 87.2 | 75.7 | 78.4 | 66.7 | 50.4 | 81.0 | 81.5 | 52.6 | 33.4 | 43.3 |

We first compare our method with other prompt-based methods. The results of our experiments show that our method achieved the best performance among prompt-based methods in 16 out of 19 datasets, significantly outperforming VPT and other VPT-based methods. Notably, CVPT achieves the highest accuracy in all datasets within the structured group, indicating that the addition of cross-attention significantly improves the adaptation of prompts. Therefore, CVPT performs better in those out-of-distribution (OOD) datasets. Additionally, since we use fewer than 20 prompts in VTAB-1K, CVPT requires the lowest number of parameters.

When considering all PEFT methods, we find that on a small dataset like VTAB-1K, almost all mainstream PEFT methods outperformed full-tuning in terms of performance. This suggests that correctly selecting the parameters to fine-tune is crucial. For our CVPT, it shows an impressive performance, which is only 0.2% behind SPT in accuracy while using fewer parameters than SPT,

and outperforms the other PEFT methods in performance. This indicates that CVPT reaches SOTA in terms of both performance and parameter count. In particular, compared to other prompt-based methods that show weaknesses, our CVPT deeply explores the potential of prompt-based methods and demonstrates that prompt-based methods can also perform well in the field of visual fine-tuning.

**FGVC.** Performance on VTAB-1K alone is not enough to prove the superiority of CVPT. Therefore, we introduce the experimental results of CVPT on FGVC to explore its performance on a complete dataset of a certain scale. The results are shown in Table.3 below:

Table 3: **Performance comparisons on five FGVC datasets with ViT-B/16 models pre-trained on ImageNet-21K.**

| datasets / Method | CUB-200 -2011 | NABirds | Oxford Flowers | Stanford Dogs | Stanford Cars | Avg. Acc. | Params. (M) |
|---|---|---|---|---|---|---|---|
| Full fine-tuning | 87.3 | 82.7 | 98.8 | 89.4 | 84.5 | 88.5 | 86.0 |
| Linear probing (14) | 85.3 | 75.9 | 97.9 | 86.2 | 51.3 | 79.3 | **0.18** |
| Adapter (18) | 87.1 | 84.3 | 98.5 | 89.8 | 68.6 | 85.7 | 0.41 |
| AdaptFormer (4) | 84.7 | 75.2 | 97.9 | 84.7 | 83.1 | 85.1 | 0.37 |
| Bias (46) | 88.4 | 84.2 | 98.8 | 91.2 | 79.4 | 88.4 | 0.28 |
| VPT-Shallow | 86.7 | 78.8 | 98.4 | 90.7 | 68.7 | 84.6 | 0.25 |
| VPT-Deep (21) | 88.5 | 84.2 | 99.0 | 90.2 | 83.6 | 89.1 | 0.85 |
| DAM-VP (20) | 87.5 | 82.1 | 99.2 | **92.3** | - | - | - |
| EXPRESS (7) | 88.3 | - | 99.0 | 90.0 | 80.5 | - | - |
| $E^2$VPT (12) | 88.5 | 84.2 | 99.0 | 90.2 | 83.6 | 89.2 | 0.45 |
| SPT-Adapter (13) | 89.1 | 83.3 | 99.2 | 91.1 | 86.2 | 89.8 | 0.41 |
| SPT-LoRA (13) | 88.6 | 83.4 | **99.5** | 91.4 | **87.3** | 90.1 | 0.48 |
| CVPT | **89.7** | **86.1** | 99.3 | 91.4 | 84.9 | **90.3** | 0.79 |

Similar to the results on VTAB-1K, our approach substantially outperforms other prompt-based methods on FGVC benchmark. Additionally, it surpasses SPT and other adapter-based methods to achieve the best performance. This suggests that CVPT exhibits better performance on relatively large datasets like FGVC, which proves the adaptability of CVPT to the increasing scale of data in the future.

**ADE20K.** Finally, we apply CVPT to SETR(49) on the ADE20K dataset to explore its performance on downstream tasks of semantic segmentation. The results are shown in Table.4 below:

Table 4: **Results of ADE20K datasets with ViT-L models.** We report "mIoU-SS" and "mIoU-Ms" which denote single-scale and multi-scale, respectively

| Methods | Params(M) | mIoU-SS | mIoU-Ms |
|---|---|---|---|
| Full-tuning | 318.3 | 48.31 | 50.07 |
| Linear probing | 13.18 | 35.12 | 37.46 |
| Bias (46) | 13.46 | 43.40 | 45.33 |
| VPT (21) | 13.43 | 42.11 | 44.06 |
| RepAdapter (30) | 13.82 | 44.44 | 46.71 |
| SPT-Adapter (13) | 14.60 | 45.20 | 47.20 |
| SPT-LoRA (13) | 14.60 | 45.40 | 47.50 |
| CVPT(P=10) | **13.43** | 43.78 | 45.85 |
| CVPT(P=200) | 18.00 | **45.66** | **47.92** |

This task is quite challenging because of the huge distribution gap between pre-training datasets and downstream tasks. In this situation, our CVPT shows a 1.7% enhancement of "mIoU-SS" over the VPT with the same number of prompts. If we use 200 prompts for fine-tuning, CVPT represents a significant improvement over the other PEFT methods. This fully demonstrates the adaptation of CVPT to OOD datasets. Besides, due to our optimization of the deployment, even though the number of learnable parameters increases by 4 million, our memory usage and training time increase by less than 10% compared to linear probing and less than 5% compared to it when using 10 prompts during training.

### 4.3 Ablation Studies

**The impact of the location of the Cross-Attention (CA).** We conducted experiments with the following five positions to explore the optimal deployment of CA, and the results of the experiments are displayed in Table.5:

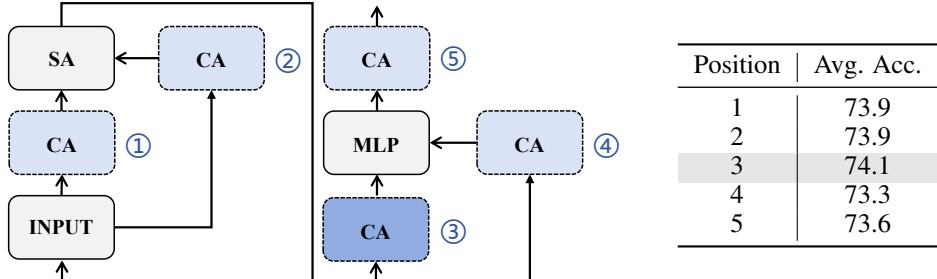

| Position | Avg. Acc. |
|----------|-----------|
| 1 | 73.9 |
| 2 | 73.9 |
| 3 | 74.1 |
| 4 | 73.3 |
| 5 | 73.6 |

Figure 5: **(a) The deployments of cross-attention in ViT.** Five possible positions can be inserted. Our final deployments are in dark blue. **(b) Performance comparisons of different deployments of cross-attention.**

We can see that inserting in prompt tokens after self-attention (SA) is the best way to perform. However, if a slight performance decrease is acceptable, we can choose position 2 to insert in parallel to improve the efficiency of the operation (this improvement is also slight).

**The impact of weight-sharing between CA and SA.** We set CA to be learnable (without weight-sharing) and frozen (with weight-sharing) respectively to investigate the impact of weight-sharing. The results on VTAB-1K and FGVC are shown in Table.5 below:

Table 5: **Performance comparisons of learnable CA and frozen CA with weight-sharing.**

| Setting | Learnable Para(M) | VTAB-1K | | | | FGVC |
|---------|-------------------|------|------|------|------|------|
| | | Nat. | Spe. | Str. | Avg. | |
| learnable CA | 28.4 | 80.1 | 84.8 | 57.8 | 74.2 | 89.4 |
| frozen CA | 0.08 | 80.1 | 84.4 | 57.8 | 74.1 | 90.3 |

We find that setting CA to tunable adds a significant number of parameters, substantially increasing computational overhead. Despite the slight performance gain it brings on VTAB-1K, it lags behind the frozen CA substantially in FGVC. Therefore, We believe that the parameters of SA are valuable for guiding the fine-tuning of CA. Especially, when dealing with a complete dataset of a certain size, such as FGVC, the weight-sharing mechanism can better utilize the pre-trained capabilities of the model, thereby improving performance.

## 5 Conclusion

In this paper, we explore the current mainstream prompt-based method VPT deeply and analyze the reasons why it performs poorly. Consequently, we propose a simple and effective PEFT method, CVPT, which introduces the cross-attention module to compute the cross-attention between the prompt tokens and embedded tokens thus instructing the model's fine-tuning. What more, the weights of cross-attention are come from self-attention, avoiding introducing an enormous number of additional trainable parameters and achieving better performance. We conducted extensive experiments on 25 datasets, and the results demonstrate that CVPT achieves SOTA performance. Additionally, we conducted extensive ablation experiments on CVPT, demonstrating the impact of introducing cross-attention and weight-sharing, as well as its efficiency and performance improvements over VPT. We hope our work will inspire prompt-based PEFT methods in the future. One limitation of our work is that CVPT does not explore new strategies for the initialization of prompt tokens. In VPT, the author made a complete comparison of different initialization methods. In our work, we take the same strategy with VPT. However, we still think the optimized specific initialization method is better than the general methods VPT used. Besides, this initialization will also help us understand how prompts help the model's fine-tuning.

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
