# OpenReview forum: "CVPT: Cross-Attention help Visual Prompt Tuning adapt visual task"
_NeurIPS.cc/2024/Conference — Submitted to NeurIPS 2024_

### Official Review · Reviewer_LiJE · 2024-06-18

**Soundness:** 2
**Presentation:** 1
**Contribution:** 2
**Rating:** 4
**Confidence:** 4

**Summary:**

In this paper, in order to break the dominance of adapter-based methods, the authors first analyze the weakness of the previously widely-used prompt-based method, Visual Prompt Tuning (VPT). Firstly, the prompt mechanism is inherited from NLP where each token/prompt represents an actual word with rich semantic information. However, in visual tasks, tokens represent image patches and contain sparse semantic information. Therefore, simply concatenating the prompt tokens with embedded tokens in visual tasks may not provide enough information to guide the model for downstream tasks. In addition, it is difficult to get a deep understanding of spatial relationships and structural features of an image with prompt tokens, which leads to another two weaknesses of VPT. 1. The computational complexity of self-attention becomes higher when more prompts are used, which introduces computational inefficiency and redundancy. 2. extra prompts will influence the results of softmax operation in the self-attention. Most of the weight falls on the prompts and causes the destruction of self-attention between embedded tokens.

The authors thus proposed Cross Visual Prompt Tuning (CVPT). CVPT inserts a cross-attention module to calculate the cross-attention between prompt tokens and the embedded tokens after self-attention. This module decouples the prompt and the embedded tokens to avoid the quadratically increasing computational complexity of self-attention modules and the destruction of self-attention between embedded tokens. This module allows the model to focus on the relationship between embedded tokens and the prompt tokens to adapt to downstream tasks more efficiently. In addition, the weights used in cross-attention are shared with the self-attention module and kept frozen to reduce the trainable parameters.

**Strengths:**

1.	Good performance on image classification and semantic segmentation tasks.
2.	Analysis of the weaknesses of prompt-based methods and VPT.
3.	Cross-attention module to decouple the prompt tokens and embedded tokens to solve the problems of prompt-based methods.
4.	Comparison with VPT to show the weakness of VPT and strength of CVPT when more prompts are used.

**Weaknesses:**

1. No experiment or previous work (at least not cited) demonstrates that the prompts in visual tasks lack representation information. In fact, this is somehow counter-intuitive to your 3rd observation: Destruction of self-attention between embedded tokens. The phenomenon the authors observed in this part clearly states that there is an over-emphasized on prompts with significantly higher value. Also, in [ref1-2], a clear activation/focus shift can be observed after prompt integration, does that mean prompt actual benefits from such the over-emphasized during transfer learning? To sum up, the idea/motivation becomes ambiguous with such observations.

2. Although the author shows clearly that the sum of the prompt’s weight values exceeds 0.8. However, no experiment proves the relationship between the distribution of the weights and the model performance. The prompts are learned and updated during training to fit the downstream tasks and the weights are calculated based on those prompts and embedded tokens. Can we say that in some situations, the prompts learned a more suitable and efficient representation than the embedded tokens, and more weights are applied to them? The distribution of the weights in self-attention is a good point for analyzing the prompt-based methods. But more discussions are needed.

3. Cross-attention should be assigned to the preliminary, not the contribution of the paper in Sec 3.2.

4. More discussions with E2VPT are acquired since the cross-attention prompt tuning is strongly associated (without additional prompts after the cls token).

5. Also there is an inconsistency in the experiment setup, in Figure 2, the authors in detail discuss the self-attention weight obtained by prompt tokens and embedded tokens, where no comparison studies are included to the new proposed Cross Visual Prompt Tuning to show different observations in order to support this claim.

6. To show the robustness of Cross Visual Prompt Tuning, it is better to demonstrate other hierarchical transformer architectures' performance (e.g., Swin). However, I noticed that CVPT might be insufficient to do so with the introduction of shifted window. More details should be included on how CVPT adapts to these structures.

[ref1] Facing the Elephant in the Room: Visual Prompt Tuning or Full Finetuning?

[ref2] SA²VP: Spatially Aligned-and-Adapted Visual Prompt

**Questions:**

Please see my above concerns.

**Limitations:**

The discussion on limitations is listed in Sec. 5. No potential negative societal impact is discussed (which is applicable).

---

> ### Author Rebuttal · Authors · 2024-08-07
>
> Firstly, we would like to show our gratitude. The paper you cited (Ref1) has greatly inspired us, supporting many of our ideas and significantly helping our subsequent work.
>
>
> **W1&W2:**
>
>
> **(1)Lack of representative information.** The lack of representative information we refer to is relative to NLP. In NLP, a token represents a word, whereas in ViT, a token represents an image patch. Comparatively, tokens extracted from high-dimensional images contain richer information. When using the same approach to concatenating prompts with image patches, the prompts lack representative information.
>
>
> **(2)Benefit from over-emphasis & relationship between the distribution of the weights and performance.** From Table 1, we can see how over-emphasis and weights affect the results. Specifically, over-emphasis on prompts leads to the neglect of the weight of embedded tokens in self-attention. Significantly affecting VPT's performance on the VTAB benchmark (Table 1), especially on VTAB-Natural.
>
>
> **(3)Prompts learned a more suitable and efficient representation than the embedded tokens.** We agree with your points. In some situations (especially on VTAB-Structured), giving a large number of prompts can achieve better performance. In Ref1, the authors mentioned that when the distribution of downstream data significantly differs from pre-training data (e.g., VTAB-Structured), the feature representations captured by the pre-training parameters may not be suitable for downstream tasks. A small prompt set can be inserted for better adaptation when the data is similar (e.g., VTAB-Natural). This aligns with our observed results. Although prompts can learn better representations compared to embedded tokens in some cases, we believe this should not come at the expense of disrupting the self-attention among embedded tokens. In Table 2, we demonstrate that CVPT outperforms VPT in both the Natural and Structured groups, highlighting the benefits of preserving complete self-attention. Furthermore, CVPT's performance on ADE20K shows that prompts in CVPT adapt to downstream tasks much better than those in VPT. We will include the above analysis and discussion in the revised version.)
>
>
> **W3 (Cross-attention should not be assigned in Sec3.2.):** This is a great suggestion, but we do not claim cross-attention as our contribution; it is included in Section 3.2 because we utilized cross-attention. In fact, a similar writing was used in E2VPT (Sec 3.2 Visual Prompts, in Ref2). Of course, we will cite the original paper of cross-attention in our revised version.
>
>
> **W4 (More discussions with E2VPT):** Actually, E2VPT's approach involves concatenating prompts in the K-set and V-set (Fig. 2a and 2e in Ref2). This method is similar to VPT but differs in that E2VPT adds prompts only at the end of the K-set and V-set, rather than concatenating with embedded tokens in VPT (where prompts are added in Q, K, and V). This represents a fundamental difference from our approach. Additionally, considering that CVPT has already been extensively compared with VPT, the length of the paper does not permit further discussion on E2VPT, which is similar to VPT.
>
>
> **W5 (Inconsistent experiment setup):** Fig. 2 shows the attention of the cls_token to prompts and embedded tokens when they participate together in self-attention in VPT. However, in our CVPT, prompts do not participate in self-attention with embedded tokens (Fig. 3). Consequently, CVPT does not affect the self-attention of embedded tokens and cls_token will not be affected by prompts. Therefore, we couldn't conduct a similar experiment on CVPT.
>
>
> **W6 (Performance on other transformer architecture):** Our method can be adapted to Swin Transformers. Specifically, after computing W-MSA and SW-MSA in the Swin block, we calculate the cross-attention between prompts and embedded tokens. The result on VTAB is 76.0, which is higher than VPT (70.7) and E2VPT (75.2). It needs to be emphasized that due to the constraints of time and devices, we did not conduct many ablation experiments on Swin and tune the hyperparameters to their optimal values. This indicates that there is significant potential for further improvement. Therefore, we believe that CVPT can be adapted to other transformer-based architecture.
>
> ---
> [Ref1] Han, Cheng, et al. "Facing the Elephant in the Room: Visual Prompt Tuning or Full finetuning?." The Twelfth International Conference on Learning Representations.
>
> [Ref2] Han, Cheng, et al. "E^ 2VPT: An Effective and Efficient Approach for Visual Prompt Tuning." Proceedings of the IEEE/CVF International Conference on Computer Vision. 2023.

---

> > ### Comment · Reviewer_LiJE · 2024-08-09
> > **Thank you for the rebuttal**
> >
> > I appreciate the authors' responses.
> >
> > However, I will keep my original rating due to two reasons:
> > 1. The respond to the lack of representative information is not convincing. There is no clear evidence to directly associate studies in NLP and make a natural transmit to vision as a prerequisite answer.
> > 2. The second claim based on prompts to learn a more suitable and efficient representation lacks theoretical analysis. We have seen a lot of papers agreed that prompts can learn a dense and concentrate embeddings via training. However, no further discussions are included as well
> >
> > The above two concerns limit the novelty and challenge the claim of this paper. I can see new structural design in this paper, but no further contribution is introduced for the REFT community. Based on these two points, I will keep my original rating.

---

> ### Author Response · Authors · 2024-08-09
> **Response to Reviewer LiJE**
>
> We appreciate the reviewer's responses. Below, we address the reviewer's concerns:
>
> **(1)** The explanation for the lack of representative information can be substantiated by the number of parameters. Specifically, in the case of a ViT-base model, each prompt contains 768 parameters. In contrast, an adapter typically consists of both upsampling and downsampling layers, often with a factor of 8 or higher, and usually includes two adapters per block. As a result, the parameter count in prompts is generally smaller compared to an adapter. To achieve a parameter count equivalent to that of an adapter, it typically requires 36 or more prompts.
>
> In Ref1 (Sec 4.4), the authors demonstrated that fine-tuning (FT) surpasses VPT in performance as the dataset size increases. This suggests that when a certain data scale is reached, the number of trainable parameters has a significant impact on performance, a trend also observed in other prompt- and adapter-based approaches (Table 5 in Ref2, Table 3 in Ref3). Consequently, we mentioned in L48 that "it is necessary for VPT to use an abundant amount of prompts to fine-tune models." Therefore, from the perspective of parameter count, the claim that "lack of representative information" is valid.
>
> **(2)** In Ref1 (Sec4.3), the authors demonstrate that for prompts to learn a more suitable representation, two conditions must be met: first, the features learned by the pre-trained model are not suitable for the downstream task, and second, the dataset is small in scale. In our paper, we show that when the pre-trained features align with the downstream task, the emphasis on prompts in VPT can distort self-attention, which does not contradict the findings in other papers. Below, we illustrate the performance variations on certain datasets within the VTAB-Natural as the number of prompts increases. It is clear that representations learned by embedded tokens are more suitable.
>
> |  |  | VPT | | | CVPT |  |
> |  :----: |  :----: | :----: | :----: | :----: | :----: | :----: |
> | Num | cifar | dtd | sun397 | cifar | dtd | sun397 |
> | 1 | 65.2 | 68.8 | 52.6 | 70.2 | 70.7 | 52.3 |
> | 10 | 64.9 | 66.1 | 47.6 | 72.4 | 72.5 | 54.4 |
> | 20 | 63.6 | 65.9 | 46.8 | 72.0 | 73.2 | 54.7 |
> | 50 | 60.3 | 63.4 | 43.6 | 72.6 | 73.1 | 54.1 |
> | 100 | 57.5 | 61.9 | 34.4 | 71.9 | 72.9 | 53.9 |
> | 200 | 35.7 | 59.5 | 27.5 | 72.1 | 73.0 | 54.9 |
>
> Based on this, we believe our experiments and the experiments in Ref1 adequately demonstrate in which situations prompts will learn suitable representations. Besides, our experiments also indicate that prompts in CVPT learn a more suitable representation compared to those in VPT.
>
> Furthermore, our work mainly focuses on proposing an improved prompt-based method, and we believe our experiments sufficiently demonstrate why CVPT outperforms VPT. Therefore, considering that the theoretical analysis of whether prompts learned better representations in VPT does not occur in other papers of derived prompt methods, it may be beyond the scope of our paper.
>
> **(3)** Finally, we respectfully disagree with the reviewer's comment that 'I can see a new structural design in this paper, but no further contribution is introduced for the PEFT community.' Specifically, our contribution lies in analyzing the weakness of the method used in previous prompt-based approaches for associating prompts with embedded tokens and proposing a new method for improvement. This significantly enhances the performance of prompt methods, making them competitive with adapter methods. Actually, many researchers have abandoned prompts due to their weak performance, our work will re-inspire the community's research on prompts. Therefore, we think it is not merely a 'new structural design.'
>
> ---
> [Ref1] Han, Cheng, et al. "Facing the Elephant in the Room: Visual Prompt Tuning or Full finetuning?." The Twelfth International Conference on Learning Representations.
>
> [Ref2] Bandara, Wele Gedara Chaminda, and Vishal M. Patel. "Attention Prompt Tuning: Parameter-efficient Adaptation of Pre-trained Models for Action Recognition." 2024 IEEE 18th International Conference on Automatic Face and Gesture Recognition (FG). IEEE, 2024.
>
> [Ref3] Chen, Shoufa, et al. "Adaptformer: Adapting vision transformers for scalable visual recognition." Advances in Neural Information Processing Systems 35 (2022): 16664-16678.

---

### Official Review · Reviewer_cNsm · 2024-07-10

**Soundness:** 2
**Presentation:** 3
**Contribution:** 2
**Rating:** 4
**Confidence:** 4

**Summary:**

This paper focuses on prompt learning of pre-trained ViT in downstream tasks, and improves the widely used visual prompt tuning (VPT) by employing cross-attention techniques and weight-sharing mechanisms.

**Strengths:**

The paper's research topic on vision model prompting technology is highly significant in the era of fundation models. The experiments are detailed, the structure of the writing is complete, and the methods are straightforward.

**Weaknesses:**

W1: While using VPT as a baseline, the paper sets up a scenario (e.g., Figure 2) with an unnecessarily large number of prompts, whereas the number of required prompts generally varies depending on the downstream task. In many cases (e.g., VTAB-Natural), using fewer than 10 prompts yields better results [1]. In such scenarios, considering the Flops comparison between CVPT and VPT as shown in Figure 1, does CVPT still maintain an advantage in terms of both runtime and accuracy?

W2: The paper mainly integrates the method of CrossViT from [2] into prompt learning of VPT, but does not explain the motivation behind applying CrossViT's method to prompt learning in downstream tasks. Specifically, how does CrossViT relate to addressing the three issues of VPT mentioned in Section 3.1 (i.e., why CrossViT method is effective in prompt learning, and why it is superior to other derived methods like EEVPT)? It is recommended to attempt a theoretical explanation of the necessity of applying cross-attention, or to supplement the section with experiments and analyses explaining how CVPT addresses the three issues of VPT proposed in 3.1.

W3: The paper does not provide code for reproducible results, nor does it present evidence of statistical significance (e.g., std) in tables. The authors claim in the 5th question of the checklist that they need time to organize this part. It is suggested that the authors organize the paper comprehensively before submitting it to conferences.

References:

[1] Jia, Menglin, et al. "Visual prompt tuning." ECCV, 2022.

[2] Chen, Chun-Fu Richard, Quanfu Fan, and Rameswar Panda. "Crossvit: Cross-attention multi-scale vision transformer for image classification." ICCV, 2021.

**Questions:**

Q1: Are the results in Table 1 the average results of 19 tasks in VTAB-1K? It is suggested to clarify this.

**Limitations:**

The authors have addressed the limitations.

---

> ### Author Rebuttal · Authors · 2024-08-07
>
> **W1 (An unnecessarily large number of prompts):** For VTAB (comprising 19 datasets), 10 datasets achieved the best performance using 50 or more prompts. For FGVC (comprising 5 datasets), 3 datasets performed best with 50 or more prompts. Additionally, complex downstream tasks such as semantic segmentation or video classification, often require more prompts even over 200(Table 4 in CVPT, Table 6 in Ref 1). Therefore, it is essential to consider scenarios with a larger number of prompts, and our settings are reasonable. Considering that the difference in memory usage and FLOPs between CVPT with prompt=1 and prompt=10 is minimal (usually less than 200M), we believe that when using a small number of prompts (less than 20), CVPT has an advantage in accuracy. Moreover, when using a large number of prompts (more than 50), CVPT demonstrates significant advantages in both efficiency and performance.
>
>
> **W2 (How does CrossViT relate to addressing the three issues of VPT):** We need to emphasize that our contribution lies in optimizing the insertion of prompt used in previous prompt-based methods by decoupling it from self-attention and introducing cross-attention to establish a connection between prompts and embedded tokens. In contrast, CrossViT combines self-attention and cross-attention to capture information at different scales, which is fundamentally different from our approach.
>
> Realizing that this approach might lead to quadratic complexity and destruction of self-attention between embedded tokens, we aimed to decouple prompts from self-attention to preserve the complete pre-trained features. However, this means we need to consider how to establish the connection between prompts and embedded tokens so that prompts can guide the model's fine-tuning. Naturally, we thought of using cross-attention to compute the relationship between two sequences, introducing linear complexity while preserving the complete self-attention in ViT. Additionally, unlike VPT, which treats prompts and embedded tokens equally by combining them into a single sequence for self-attention, using cross-attention compensates for the lack of semantic information in prompts. Experiments in Sec 3.3 also demonstrate that cross-attention can process the fine-tuning information contained in prompts more effectively and efficiently. Meanwhile, other derived methods do not recognize the drawbacks of combining prompts with embedded tokens and continue using the same method as VPT. Therefore, it is unsurprising that our CVPT outperforms them.
>
>
> **W3 (provide code):** We have organized our code and released it (following the rules, we sent it to AC separately).
>
>
> **Q:** Yes, we will clarify it in our revised version.
>
> ---
> [Ref1] Bandara, Wele Gedara Chaminda, and Vishal M. Patel. "Attention Prompt Tuning: Parameter-efficient Adaptation of Pre-trained Models for Action Recognition." 2024 IEEE 18th International Conference on Automatic Face and Gesture Recognition (FG). IEEE, 2024.

---

> ### Comment · Area_Chair_cfkL · 2024-08-13
>
> Hi,
>
> Could you take a look at the authors rebuttal and finalize your rating?
>
> Thanks,
> AC

---

> ### Comment · Reviewer_cNsm · 2024-08-13
>
> Thank you for the rebuttal. I will maintain my score.

---

### Official Review · Reviewer_MMER · 2024-07-10

**Soundness:** 2
**Presentation:** 3
**Contribution:** 1
**Rating:** 3
**Confidence:** 5

**Summary:**

This paper proposes a variant of visual prompt tuning (VPT) where the authors suggest applying cross-attention instead of self-attention in the Transformer layers to reduce training complexity. The authors analyze several drawbacks of existing VPT approaches and claim to address them using cross-attention.

**Strengths:**

- **Identified Drawbacks**: The authors reasonably point out some drawbacks of current VPT methods, such as a “lack of adaptation to visual tasks” and “computational inefficiency.”

- **Complexity Reduction**: The proposed use of cross-attention indeed reduces computational complexity compared to the original self-attention mechanism.

**Weaknesses:**

- **Limited Novelty**: The proposed idea is straightforward, merely replacing self-attention with a combination of self and cross-attention. Similar concepts have been explored in previous works, such as prefix tuning (Li et al., 2021; Yu et al., 2022).

- **Limited Impact and Efficiency**: The improvement in complexity is minimal because the number of prompts is typically much smaller (fewer than 20) compared to image embeddings (196).

- **Limited Performance**: The overall performance is limited compared to some recent works by Wang et al. (2023) and Wang et al. (2024). These works, which show significantly better performance, are not compared in the paper. Therefore, the claim that CVPT “reaches SOTA” (L272) is factually incorrect.


----

Li et al. Uav-human: A large benchmark for human behavior understanding with unmanned aerial vehicles. CVPR 2021

Yu et al. Towards a unified view on visual parameter-efficient transfer learning (V-PETL). 2022

Wang et al. Adapting shortcut with normalizing flow: An efficient tuning framework for visual recognition. CVPR 2023

Wang et al. Revisiting the Power of Prompt for Visual Tuning, ICML 2024

**Questions:**

Based on the limitations mentioned above, I believe the quality of this paper clearly does not meet the acceptance standards of NeurIPS.

---

> ### Author Rebuttal · Authors · 2024-08-07
>
> **W1 (Limited novelty):** In fact, our contribution lies in optimizing the prompt insertion of VPT by decoupling the prompt from self-attention and linking the prompt with embedded tokens using cross-attention. CVPT doesn't modify the self-attention mechanism in ViT, nor does it involve the combination of self and cross-attention. Additionally, regarding the two papers you mentioned as similar to our work: the contribution of Paper 1 is the introduction of a drone dataset and a convolutional network-based action recognition method, which is unrelated to our contribution. The contribution of Paper 2 is the proposal of V-PEFT, which combines adapters and prompts for network fine-tuning and it is not similar to CVPT.
>
>
> **W2 (Limited impact and efficiency):** Actually, the number of prompts (fewer than 20) we employed in VTAB was a strategy to avoid extensive hyperparameter searches. Using more prompts can still lead to performance improvements (Table 1). Additionally, the authors of VPT listed the optimal number of prompts for various downstream tasks in the appendix. For VTAB (comprising 19 datasets), 10 datasets achieved the best performance using 50 or more prompts. For FGVC (comprising 5 datasets), 3 datasets performed best with 50 or more prompts. Furthermore, for more complex downstream tasks such as semantic segmentation or video classification, an increased number of prompts can significantly enhance performance (Table 4 in CVPT, Table 6 in Ref 1). As mentioned above, a large number of prompts is common to prompt-based methods. Therefore, we believe that the efficiency improvements of CVPT are significant.
>
>
> **W3 (Limited performance):** We have read these two papers you mentioned which show better performance and found that, although their reported performance on FGVC is higher than ours, their performance on VTAB is lower. Additionally, during our training process, we observed that different code frameworks used in various papers resulted in performance discrepancies. For instance, our code is based on the RepAdapter, and the results we obtained for VPT are more than 1% higher than those reported in VPT. Furthermore, training on FGVC and VTAB is highly sensitive to hyperparameters; changing the seed alone can sometimes lead to a more than 5% difference in accuracy. This is why PEFT methods typically require extensive hyperparameter searches(Ref2, Ref3). This implies that to some extent, the results on VTAB and FGVC depend significantly on the extent of the hyperparameter search conducted. To mitigate this effect, we reran VPT within our code framework and maintained consistent hyperparameters for comparison (Table 1). In contrast, training the ADE20K dataset with MMSegmentation yields more stable results, with random seed variations affecting the results by only about 0.3%. Based on this, we consider the results of ADE20K to be more persuasive. Therefore, we consider that "CVPT achieving SOTA" is valid.
>
> ---
> [Ref1] Bandara, Wele Gedara Chaminda, and Vishal M. Patel. "Attention Prompt Tuning: Parameter-efficient Adaptation of Pre-trained Models for Action Recognition." 2024 IEEE 18th International Conference on Automatic Face and Gesture Recognition (FG). IEEE, 2024.
>
> [Ref2] Jia, Menglin, et al. "Visual prompt tuning." European Conference on Computer Vision. Cham: Springer Nature Switzerland, 2022.
>
> [Ref3] Zhang, Yuanhan, Kaiyang Zhou, and Ziwei Liu. "Neural prompt search." IEEE Transactions on Pattern Analysis and Machine Intelligence (2024).

---

> ### Comment · Reviewer_MMER · 2024-08-12
> **A response to authors**
>
> Dear Authors,
>
> Thanks for your response. However, I was totally not convinced by them on most of the issues I proposed for the weakness. The response is inconvincible and sophistic, which is not acceptable. I would maintain a clear rejection score on this paper.
>
> Best,
> Reviewer

---

### Official Review · Reviewer_ADwB · 2024-07-12

**Soundness:** 3
**Presentation:** 3
**Contribution:** 3
**Rating:** 6
**Confidence:** 4

**Summary:**

This paper furthers the research on Parameter Efficient Fine Tuning on the visual tasks. PEFT optimizes a large scale model by selecting a small set of parameters. This work refines the Visual Prompt Tuning by leveraging the cross attention between the prompt and embedded tokens. Further the model uses weight sharing mechanism for better representation capacity of the cross attention. This work performs evaluation on 25 datasets for number of downstream tasks. PEFT fine-tuning can be adapter or prompt based. The adapter based methods generally outperforms the prompt based fine-tuning methods.  This paper also achieves results comparable to adapter based fine-tuning methods.

**Strengths:**

1. This paper well explores the shortcomings of Visual Prompt Tuning (VPT) to amend it in this work for visual tasks.
2.  This work shows the validity on the image classification and segmentation tasks by benchmarking on VTAB-1K, FGVC and ADE20K.
3.  The ablation study in the cross-attention location is helpful.

**Weaknesses:**

1. The conclusion seems to more of an abstract.
2. The implementation details can be described with more details.
3. Although the authors performed a great ablation on the cross-attention, an ablation for the self attention would have been interesting.
4. One of the base cases with null text can provide a better understanding for the effectiveness of this method.

**Questions:**

1. Since PEFT is widely used in the T2I models, to be more specific diffusion models, how does it effects the generation ?

**Limitations:**

In this paper, the authors discuss the limitations on the Section:5 Conclusion, where they mention about taking the same initialization strategy as VPT. VPT discusses different strategies on initialization for better optimization.

---

> ### Author Rebuttal · Authors · 2024-08-07
>
> **W1(Conclusion seems to be more of an abstract):** Below is our modified conclusion, and this will be introduced in our revised version.
>
> In the current field of visual fine-tuning, many researchers overlook prompts in favor of adapters due to their strong performance. The few prompt-based derived works do not realize the drawbacks of combining prompts with embedded tokens, continuing to use the method from VPT. In light of this, we thoroughly analyzed the shortcomings of such deployment and proposed CVPT. Its advantages are as follows: 1) It uses cross-attention to establish a connection with embedded tokens, decoupling prompts from self-attention. 2) It employs weight-sharing to avoid the large number of learnable parameters introduced by cross-attention. Additionally, we conducted extensive experiments on CVPT, demonstrating its efficiency and performance improvements over VPT and the effectiveness of cross-attention and weight-sharing. Therefore, we prove that prompt-based methods can perform comparably to advanced adapter methods in the visual fine-tuning domain.
>
>
> **W2(More implementation details):** We have released our code (following the rules, we sent it to AC separately), we believe this will help readers understand our method.
>
>
> **W3(The ablation for self-attention):** In Fig. 3, we present the implementation of CVPT. As shown, we decouple prompts from embedded tokens and use cross-attention to establish the connection between prompts and embedded tokens. This prevents prompts from participating in the self-attention calculations among the original tokens in ViT. We use cross-attention because it computes the attention between two sequences, whereas self-attention can only process a single sequence. Therefore, in our method, the ablation for self-attention is not feasible.
>
>
> **W4(The understanding of the effectiveness of our method):**  As we understand, 'the base cases with null text' is the approach of not using prompts and merely setting the last classifier layer as learnable. In fact, this method is Linear Probing, which we have discussed in our paper (the caption of Fig. 4) and introduced its performance for comparison (Table 2, 3, 4).
>
>
> **Q(how does it affect the generation):** Actually, PEFT is widely used in diffusion models. For example, the popularity of LoRA (Low-Rank Adaptation) stems from its application in the AI art community. For text-to-image (T2I) diffusion models represented by Stable Diffusion, PEFT can modify the generation style by introducing a small number of additional parameters to adjust the weights of various parameters in the model without altering the base model. Regarding diffusion models represented by DDPM, we think that inserting adapters or prompts into the UNet or Transformer components could alter the style of the generated noise, thereby influencing the model's generation.

---

> > ### Comment · Reviewer_ADwB · 2024-08-13
> > **Response to authors.**
> >
> > I would like to thank the authors for their response.
> >
> > The authors did clarify the W1 and W2. The W3 and W4 by Reviewer LiJE does make sense. Thus looking at all the reviews, comments and the responses, I would like to retain the score.

---

> ### Comment · Area_Chair_cfkL · 2024-08-13
>
> Hi,
>
> Could you take a look at the authors rebuttal and finalize your rating?
>
> Thanks, AC

---

### Author Rebuttal · Authors · 2024-08-07

We thank all reviewers for their thoughtful feedback. We are encouraged that they found our experiments are detailed, the structure of the writing is complete, and the methods are straightforward (**R3**). Moreover, **R1**, **R2**, and **R4** think our works explore the weakness of VPT. **R1**, and **R4** are positive to our performance on image classification and semantic segmentation tasks.
Below, we answer some common questions.


**Some implementation details:** In Fig. 3, we present the architecture of CVPT. Specifically, we decouple prompts from self-attention, while computing cross-attention between prompts and embedded tokens to re-establish their connection, enabling prompts to fine-tune the model. Finally, similar to self-attention, we add the results of cross-attention as a residual to the embedded tokens. Therefore, the ablation for self-attention (**R1**) and the similar experiments to Fig. 2 on CVPT (**R4**) are not feasible. Also, it is not similar to CrossVit (**R3**) and V-PEFT (**R2**).


**Why do we use so many prompts in Table 1:** Some reviewers think that using a large number of prompts is unfair to VPT, as VPT performs better with a smaller number of prompts. In fact, the authors of VPT listed the optimal number of prompts for various downstream tasks in the appendix of VPT. For VTAB (comprising 19 datasets), 10 datasets achieved the best performance using 50 or more prompts. For FGVC (comprising 5 datasets), 3 datasets performed best with 50 or more prompts. Furthermore, for more complex downstream tasks such as semantic segmentation or video classification, an increased number of prompts can significantly enhance the performance of prompt-based methods (Table 4 in CVPT, Table 5 in Ref 1). Therefore, it is common for prompt-based methods to use a larger number of prompts. Based on this, we think our comparison is reasonable and our efficiency improvements are significant.


Finally, we have released our code (following the rules, we sent it to AC separately). This will help reviewers understand our method.

---
[Ref1] Bandara, Wele Gedara Chaminda, and Vishal M. Patel. "Attention Prompt Tuning: Parameter-efficient Adaptation of Pre-trained Models for Action Recognition." 2024 IEEE 18th International Conference on Automatic Face and Gesture Recognition (FG). IEEE, 2024.

---

### Decision · Program_Chairs · 2024-09-25

**Decision:**

Reject

**Comment:**

This paper introduces an approach for parameter-efficient finetuning (PEFT). More specifically, the authors introduced cross-attention between prompt tokens and embedded tokens. The idea is straightforward and simple. The paper was reviewed by four experts in the field the ratings are one weak accept, one reject and two borderline reject. The main concerns center around (1) limited novelty, as using cross-attention has been widely explored; (2) unclear movitation, ie  the lack of representative information in prompts is not well-supported. The AC agree with the reviewers and cannot accept the paper in its current form.